# Tumor Hypoxia and Circulating Tumor Cells

**DOI:** 10.3390/ijms21249592

**Published:** 2020-12-16

**Authors:** Walter Tinganelli, Marco Durante

**Affiliations:** 1Biophysics Department, GSI Helmholtzzentrum für Schwerionenforschung GmbH, 64291 Darmstadt, Germany; w.tinganelli@gsi.de; 2Institut für Festkörperphysik, Technische Universität Darmstadt, 64291 Darmstadt, Germany

**Keywords:** CTCs, DTCs, EMT, metastasis, invasion, migration

## Abstract

Circulating tumor cells (CTCs) are a rare tumor cell subpopulation induced and selected by the tumor microenvironment’s extreme conditions. Under hypoxia and starvation, these aggressive and invasive cells are able to invade the lymphatic and circulatory systems. Escaping from the primary tumor, CTCs enter into the bloodstream to form metastatic deposits or re-establish themselves in cancer’s primary site. Although radiotherapy is widely used to cure solid malignancies, it can promote metastasis. Radiation can disrupt the primary tumor vasculature, increasing the dissemination of CTCs. Radiation also induces epithelial–mesenchymal transition (EMT) and eliminates suppressive signaling, causing the proliferation of existent, but previously dormant, disseminated tumor cells (DTCs). In this review, we collect the results and evidence underlying the molecular mechanisms of CTCs and DTCs and the effects of radiation and hypoxia in developing these cells.

## 1. Introduction

Radiotherapy remains the least invasive method to treat cancer. Although local control in early-stage malignancies easily exceeds 90%, overall survival remains dismal in many cancers due to distal metastasis. About 90% of cancer deaths are due to metastases [1]. Usually, metastases occur only at an advanced stage of cancer, and if diagnosed in time, most cancers can be cured [2].

Cancer cells with unregulated growth, metabolism, and lack of contact inhibition create an incomplete and inefficient tumor architecture that generates large areas where oxygen and nutrients are scarce. As a result of these stressors, some cancer cells acquire genetic and epigenetic alterations. They go through reversible phenotype changes that promote migration and invasion into circulatory systems [3,4,5,6,7]. These cells undergo a process that is called epithelial–mesenchymal transition (EMT). Hypoxia plays a key role in the activation of the EMT [2]. However, although all cells in which EMT is activated experience increased invasiveness and migration potential, not all of them can generate metastases. EMT cells can still intravasate, a process that leads cells to invade blood or lymphatic vessels, but they cannot survive the blood or lymphatic flow and are not then ready to generate metastases (Figure 1). Only a specific subpopulation undergoing this phenotypical transition can induce metastasis, i.e., the circulating tumor cells (CTCs).

CTCs detach from the primary tumor, invade the basal lamina, migrate to the blood vessels wall, and after intravasation, circulate in the bloodstream until they reach a site to colonize [8]. As compared with other cells subject to EMT, CTCs acquire additional characteristics, which make them particularly aggressive and resistant to chemo- and radiotherapy. This resistance is due to an accumulation of gene alterations by somatic mutations. Changes in CTCs include resistance to the programmed cell death pathway specific to detached cells (anoikis) [9].

The phenotypic transition process from epithelial to mesenchymal must be reversible to allow the cells to extravasate and colonize new niches and form metastases. CTCs, therefore, abandon their mesenchymal characteristics and go through a mesenchymal–epithelial transition (MET). The transition process is slow, follows several gene expression modifications, and goes through fundamental biological pathways. Jie et al. demonstrated that an intermediate phenotype in which the cells had not totally lost their epithelial characteristics and had not yet reached a complete mesenchymal phenotype was the most aggressive phenotype [10].

This hybrid phenotype allows the cells to detach from the primary tumor and quickly colonize a new site. There are two different models of how the colonization process occurs. In the EMT/MET model, the CTCs change their phenotype from epithelial to mesenchymal and from mesenchymal to epithelial to restore their epithelial properties when they reach a colonization site. In the collective migration model, the circulating cells move in big clusters of cells with different grades of EMT phenotypes. In these clusters, it is possible to find cells with a complete epithelial phenotype, cells with a full mesenchymal phenotype, and cells with a hybrid, half-mesenchymal and half-epithelial phenotype. In this case, the mesenchymal cells would invade the target organ and prepare the way for the epithelial cells that will be ready, in the cluster, to start the colonization. Those two models, in principle, could also coexist [10].

CTCs are highly heterogeneous. CTCs consist of epithelial cells (high-EpCAM), low-EpCAM cells, EMT cells, circulating tumor stem cells (CTSCs), and hybrid EMT/MET cells [11]. The epithelial cell adhesion molecule (EpCAM) is used to isolate a subpopulation of CTCs with a high-throughput technology [11]. Those CTCs characterized by the high EpCAM expression result in a poor prognosis [12]. CTCs have irregular shapes, bigger dimensions, and different subcellular morphology from the other bloodstream cells. Their concentration is only 1–10 cells per ml of blood in metastatic patients [13]. Although it is a very complicated procedure, it is still possible to isolate these cells from the blood. CTC blood isolation has become a non-invasive method for studying tumor stage and tumor cell characteristics and markers [14]. Furthermore, the identification and analysis of CTCs can become an excellent weapon against the tumor itself.

## 2. Circulating Tumor Cell (CTC) Characteristics

The bulk of cancer cells create incomplete and chaotic tumor architecture. In this chaos, blind blood vessels, and fast cell growth, produce an acidified and hypoxic microenvironment. Those stressors shape and select a subpopulation of aggressive and invasive cells [3,7]. Cancer cells quickly readjust their metabolism and begin to use alternative energy sources such as glucose, glutamine, pyruvate, lactate, and lipids. These cells adapt to such an extent that they use the remains of other necrotic cells or their lipids via macropinocytosis [3]. Still, through the metabolic symbiosis, hungry cells emit “feed me” signals and command other cells in the vicinity to release amino acids, fatty acids, etc. The chronic exposure to stresses provokes the transition from benign tumors, with non-invasive cells, to malignant tumors [5].

Hypoxia and starvation play a crucial role in CTCs formation. Low nutrient/oxygen supply within the tumor promotes cell invasion [15]. This transition is characterized by the cell acquisition of genetic mutations or epigenetic changes that increase the survival probability and increase the cells’ invasive phenotype. The Neural Wiskott–Aldrich Syndrome protein (N-WASP) is specifically crucial for the formation of invadopodia, actin-rich protrusions of the plasma membrane which are associated with extracellular matrix degradation. Inhibition of N-WASP with shRNA sharply decreased the metastases formation and the number of CTCs in a mouse and rat model of breast cancer [16].

Cells from the primary tumor invade the surrounding stroma gradient, driven by nutrient and growth factor. Once they reach the blood vessels, they intravasate, penetrating the vessel wall. Due to inadequate regulation of the tumor angiogenic signaling, including vascular endothelial growth factors (VEGF), vascular dysfunction contributes to increasing vessel permeability and fragility and ending in cell accessibility [17]. The inflammatory leukocytes, attracted by specific growth factors and cytokines, can facilitate intravasation [18]. Matrix metallopeptidases (MMPs) produced by these cells play an essential role. Two important MMPs delivered by infiltrating leukocytes are MMP-9 by neutrophils and MMP-13 by monocytes/macrophages [19]. Furthermore, they can provide pro-angiogenic factors and provide the means of dissemination through leaky, newly developed blood vessels [20].

PHD2 or hypoxia-inducible factor prolyl hydroxylase 2 (HIF-PH2) is an oxygen sensing molecule that targets the hypoxia inducible factor (HIF) transcription factor for its degradation. HIFs undergo a polyubiquitination signal catalyzed by the E3 ubiquitin-protein ligase that drives the molecule to proteasomal degradation in normal oxygen conditions. This is due to the hydroxylation of proline 402 or 564 by PDH2. Under hypoxic conditions, hydroxylation is inhibited, and HIF-1α rapidly accumulates [2]. PHD2+/− heterozygous deficient mice injected with PHD2+/+ tumor cells formed tumors but with low-grade intravasation and metastases. When the same cells were injected directly into the bloodstream, they readily form metastatic lesions. These results show that the vasculature’s genetic, oxygen concentration, and the PHD2 are all fundamental players in the CTCs formation [21,22].

The reduced activity of the HIF inhibitor and Von Hippel–Lindau (VHL) molecule following a lack of oxygen, and then the stabilization of the HIF1α, HIF1β, and HIF2, mediate an adaptive response in which the cells become more resistant to any treatment, activate a metabolic reprogramming, regulate the microenvironmental pH, and increase the VEGF expression.

Cancer cells increase the SNAI1 and TWIST expression, and then lose their epithelial markers, such as E-cadherin, epithelial cell adhesion molecule (EpCAM), and acquire through the expression of vimentin and N-cadherin, mesenchymal markers [5]. However, cytokeratins, such as 8 and 19, are well established for CTCs and DTCs [23]. Cancer cells can then break the basement membrane to enter into the blood vessels and become CTCs. The extracellular matrix density, and different extracellular molecules such as TGFβ, epidermal growth factor (EGF), hepatocyte growth factor (HGF), insulin-like growth factor (IGF), fibroblast growth factor (FGF), Notch and homologous wingless (wg), and Int-1 (Wnt) protein family are essential to force cancer cells through the invasive phenotype [24].

Generally, cancer cells that detach from the tumor bulk lose the adhesion and enter the circulation are slated to die according to a programmed death, anoikis [25]. However, CTCs gain the ability to survive anoikis [25]. Anoikis is compromised during the oncogenic epithelial-to-mesenchymal transition (EMT). The NRAGE (neurotrophin receptor-interacting melanoma antigen) molecule interacts with a component of the E-cadherin complex, ankyrin-G, and in doing so, remains in the cytoplasm. Oncogenic EMT downregulates ankyrin-G, enhancing the nuclear localization of NRAGE. The oncogenic transcriptional repressor protein TBX2 interacting with NRAGE represses the tumor suppressor gene p14ARF. This gene’s product, the P14ARF molecule, is a vital molecule to sensitize the cells to the anoikis (Figure 1). Overcoming anoikis is crucial in a series of changes that a tumor cell undergoes during malignant transformation [26].

Not all the CTCs can, therefore, survive the bloodstream or the lymphatic circulation. These CTCs fluctuate as single or cluster cells, and they show a genotype and phenotype very different from each other [7]. Cancer cells exposed at low oxygen pressure, and then to the bloodstream, acquire a “hypoxic memory or genetic signature”. This memory is preserved even after the cells are reoxygenated and, for many years, is stored in the disseminated tumor cell (DTC) [16]. The extravasation process can be active or passive. Cell clusters may remain stuck in small capillaries and exit through damaged blood vessels or interact with E-selectin expressed on the endothelial cells [7]. For the extravasation, CTCs need to increase FGF2b and E-cadherin and decrease fibronectin and vimentin enormously [7].

The selectine, interacting with the CTCs, slows them down first until they are blocked. Then, the CTCs can form stronger bonds, more stable interactions with other molecules such as CD44, MUC1, and integrins [7]. Furthermore, the production of TGF-β from tumor cells or platelets can also facilitate the opening of tight endothelial junctions allowing CTCs to cross blood vessels more easily.

In addition to the vasculature and the microenvironment, the immune system also plays a role in CTC formation. Macrophages can also regulate the generation of CTCs and help them for the intravasation [27]. Tumor-associated macrophages (TAM) are responsible for creating a favorable CTCs microenvironment [28,29]. In hepatocellular carcinoma (HCC), the IL-8 cytokine stimulates the M2 TAM phenotype formation. The M2 macrophages are notorious because they have a robust immunosuppressive capacity [29], enhancing tumor growth. Furthermore, they also promote EMT enormously, increasing cancer cells’ invasive potential [28]. Mechanistic studies have revealed that TAMs induce EMT in cancer cells by regulating the STAT3/miR-506-3p/FoxQ1 pathway [30]. TAMs also secrete the chemokine (C-C motif) ligand 2 (CCL2) that, in turn, increases the inflammatory response [30]. CCL2 indeed promotes macrophage recruitment. These results demonstrate a positive feedback loop between tumor cells and TAM, which promotes metastases by enhancing the CTCs through the EMT program [30].

Similar results have also been found in pancreatic cancer [31,32,33].

The granulocyte-macrophage colony-stimulating factor (GM-CSFs) recruits other macrophages and, when they reach their destination, they assume the M2 phenotype [34]. Furthermore, the Tie2-expressing macrophages (TEM), well known for promoting angiogenesis, can also promote metastasis and are often found in perivascular locations.

The VEGFA produced by these macrophages leads to transient vascular permeability, loss of vascular junctions, and increased intravasation locally at sites where tumor cells, macrophages, and blood vessels are nearby [34]. This would allow CTCs easier access to the bloodstream. However, the immune system would typically attack the CTCs once in the bloodstream. However, CTCs can evade immune surveillance. CTCs do not express the common leukocyte antigen CD45 but still can adhere to platelets and immune cells such as CD45-positive cells. Liu and his group hypothesized that CTCs could be covered by the myeloid-derived suppressor cells (MDSC), which accumulate during cancer progression and are well known for enormously increasing the metastatic rate [35] and the immunosuppressive response. CTCs, in this way, can become invisible to immune surveillance [35].

## 3. Single and Cluster CTCs

New and recent technologies have allowed the most in-depth study of CTCs. The isolation of CTCs from peripheral blood is a very complex operation. However, thanks to new techniques, they are more accessible every day, and many discoveries have been made in recent years.

It has been discovered that CTCs do not always migrate as single cells, but in some cases, they fluctuate in clusters of several cells (from tow to less than 100 cells) [36]. There is a significant difference in their ability to form metastases of single and clusters circulating cells. While most individual CTCs are cancer cells that have gone through the EMT process and cannot survive in the bloodstream, cell clusters have also undergone a metabolic shift that has made them similar to the most common cancer stem cells (CSCs) [36]. These cells have the characteristic, just like cancer stem cells, to grow in spherical structures when cultivated in vitro [37]. This is caused by increased cell surface production of the transmembrane molecule CD44, a transmembrane glycoprotein used to isolate breast cancer stem cells [36], and the molecule plakoglobin [36]. In mouse experiments, silencing the plakoglobin gene eliminated the formation of clusters by reducing the number of metastases in animals by 30–40 times [36]. Goto et al. recorded CTC clusters from the primary tumor detachment phase to the bloodstream passage, and then to the micrometastases formation, in a genetically modified lung cancer mouse model. The results showed that the clusters’ metastatic potential increased when a subset of these cells overexpressed the keratin-14 [36].

The binding sites for stemness-associated transcription factors are hypomethylated into the cells that compose the cell clusters, including binding sites for OCT4, NANOG, SOX2, and SIN3A, paralleling embryonic stem cell biology [38]. Studies in prostate cancer patients showed an average survival rate of eight times higher in patients where single circulating cells and not clusters were isolated from blood. Similar studies have been conducted on colon cancer and other cancer types and showed similar results [39], however, the result was unexpected. In fact, it was thought that the clusters of cells, having a certain steric encumbrance, could not reach the small capillaries, and therefore, this would have been a sort of selection in favor of small clusters or even single cells. That is ostensibly not the case. To study the behavior of CTCs during the passage in small vessels, microfluidic constriction devices, simulating artificial microvessels, were used [40].

Big cell clusters have been demonstrated to transit through microchannels of 50 to 300 µm diameter [40]. Clusters of circulating cells were able to unroll, deactivating the cellular connections, becoming a chain of cells that could infiltrate the capillary, and then reformed a cluster once at destination [40]. Cluster CTCs could unfold when entering constrictions and, immediately after, retract in their rounded morphology, The morphological changes involved the clusters of cells and also the cells themselves. During the squeezes, the cell nuclei could deform from rounded to elongated ellipsoidal morphologies [40].

This peculiar feature provides these clusters with significant metastatic potential [40]. Therefore, it is clear that CTCs cannot be considered to be like a single general subpopulation of cells, but rather, a very heterogeneous cell subpopulation. Single circulating cells, clusters of circulating cells, and circulating cells with stem-like characteristics [41] are all part of the same family of aggressive cells that derives from individual primary tumors [13] and they do not aggregate in the bloodstream.

In a 2014 study, the team of Aceto [13] prepared solutions of 200,000 LM2 cells (MDA-MB-231-LM2 (LM2) cells, a lung-metastatic variant of MDA-MB-231 human breast cancer cells) modified to express GFP, as single cells or in clusters. He injected these cells into the tail vein of immunodeficient mice. Although both cell populations could reach the animal’s lung, the signal for individual cells decreased rapidly in the following days due to massive apoptosis, demonstrated by staining for cleaved caspase 3. In contrast, the clusters persisted over time, indicating a greater ability to resist apoptosis and greater expansion speed. Lung tumors developed anyway for both single cells and clusters. Still, mice injected with cell clusters had a lower survival as compared with mice injected with single cells, with 12.7 weeks for LM2-CL (clusters) versus 15.7 weeks for LM2-SC (single cells) [13]. Aceto’s studies confirmed that the CTCs clusters were approximately 50 times more likely to give rise to a metastatic deposit than a single circulating tumor cell. In an orthotopic mouse model of breast cancer, despite the cell clusters being much rarer than single circulating cells, they contributed equally to the lung’s metastatic burden. After injection, the CTCs clusters persisted for less time in blood vessels, only 6–10 min, as compared wjith the single cells circulating for about 30 min [13].

## 4. CTCs, Disseminated Tumor Cells (DTCs), and Dormancy

CTCs communicate through a complex cell-cell communication and through the release of exosomes, coordinating the expression of specific cell surface molecules, such as integrins [42], to metastasize specific niches away from the primary tumor. Circulating cells, especially clusters of circulating cells, can be blocked in the small blood vessels forming small thrombus. Thanks to that, the fibronectin can activate and interact with the tumor cell integrins [42]. This gives the cells the signal to extravasate. The integrins can interact with the ECM in the proximity of the blood vessels and dictate if the seeded cells, now called disseminated tumor cells or DTC, should continue to proliferate or should become dormant. The resident cells also interact with these integrins, activating the promigratory and proinflammatory S100 gene [43]. The S100 molecule is involved in regulating protein phosphorylation, cell growth, differentiation, and inflammatory response. It plays a crucial role in cytoskeleton dynamics, by preparing the premetastatic site and promoting CTCs homing in the metastatic niche [43]. Different integrin classes give the cells a specific ability to colonize specific tissue niches, and each cancer shows a particular pattern of metastatic colonization [44,45]. The expression of integrins such as the α6β4 and αvβ5, for example, is associated with lung and liver metastases [43].

Sometimes, it is the primary tumor itself that can create a pre-metastatic niche, inducing tissues remodeling [43]. In colorectal cancer, cells secrete VEGFA that stimulates TAMs to produce the chemokine (C-X-C motif) ligand 1 (CXCL1). This molecule attracts the MDSC in the pre-metastatic liver niche. Those cells finally prepare the arrival of the CTCs and the metastasis formation [46].

The wound-like milieu accelerates the metastasis formation [47], converging CTCs. The CTCs can reengage additional platelets facilitating tumor cell adhesion, extravasation, and in the end, metastasis. This is amplified by the IL6 that is overproduced by hypercoagulable cancers, in which tumors can activate the coagulation system. Indeed, IL6 stimulates thrombopoietin production increasing the number of circulating platelets [48].

Once the CTCs reach their destination, they can go through a proliferating or dormant/hibernating G0/G1 state. It is not clear yet which signal is necessary for the cells to decide whether to proliferate or remain dormant as DTCs, and it is not known what gives the DTCs the order to start to proliferate again and form metastasis. TGFβ1 and periostin (POSTN) could be critical players, promoting the dormancy exit [49]. But dormancy itself can be considered to be a mechanism of two different co-existing factors. It can be referred to as a cell level dormancy, tumor cells dormancy, in which tumor cells exit from their normal cell cycle and stay in a prolonged G0/G1 state [50]. In this case, those cells have a lack of growth-promoting signals or downregulation of genes that affect growth at secondary sites. Among those, some of the essential genes that have been identified in this process are KISS1, MKK4, MKK6, BHLHLB3/Sharp-1, and Nm23-H1 [50].

The DTCs need to adapt to the new microenvironment. Their metabolism is deficient, similar to if they were hibernated. The high activity of the p38 pathway induces overexpression of the NF-kB transcription factor [50]. The p38 is essential for regulating cell proliferation, activating G1/S and G2/M checkpoints. Furthermore, p38 decreases the amount of cyclin D1 molecule [50], fundamental for the G1/S phase transition. Usually, cyclin D1 accumulates in the nucleus rapidly and decreased when the cells enter the S phase [51]. The production of ligands, such as the growth arrest-specific protein 6 (GAS6) for the activation of AXL receptor Tyrosine kinase signaling, as well as bone morphogenetic protein 7 and 4, activates the p38 that, in turn, stimulates the expression of the cell cycle inhibitors and the transforming factor β2 (TGFβ2) for the expression of the cell cycle inhibitor p27 [51]. 

The MKK4 and MKK6 molecules are the upstream activators of p38, and the BHLHB3 is specific for the quiescence induction, being the target of the p38 [50]. In any case, the mechanisms that are involved in the dormancy of the DTCs are different. Furthermore, tumor dormancy could also be referred to as a superior level, in which it is the tumor itself is dormant. In this case, the cell depleting phenomena, such as apoptosis or immunosurveillance at the tumor population level, induces dormancy [52].

Cancer cell metastasis can occult itself, remaining in a state of micrometastasis. That small micrometastasis can proliferate continuously but always keep a small number of cells due to the counterbalancing effects of a high apoptotic rate [52]. DTCs strongly downregulate the MHC class I molecules, becoming invisible to the cytotoxic T-cells and, more generally, by the immune system, immunological escape.

The phenotype of DTCs can give a lot of information about the evolution of some types of tumors. Most patients in which DTCs have a downregulated urokinase receptor (u-PAR) are associated with lower clinical tumor recurrence and tumor-related death [53]. DTCs can survive with severely reduced metabolism [51], and they are very resistant to any treatment because they are not proliferating. However, the mechanisms at the base of this phenomenon are much more complicated. The immune system is necessary for the protection and spread of CTCs, but it plays a critical role in preserving DTCs and their quiescent state [54]. Regulatory T cells (Treg) help promote DTCs. They release adenosine that, in turn, protects the quiescent cells from oxidative stress [50]. Furthermore, dormant cells are protected from the CD8+ T cells while killing the cycling DTCs by the cytotoxic effects [54]. 

In the DTCs, the p38 upregulates the ER chaperone BiP/Grp78 that inhibits Bax. This gives to the DTCs a strong chemoresistance. The DTCs overexpress the ATF6 molecule that activates the GTPase, and then the mTOR, the S6K, and the S6RP phosphorylation molecules [50].

There are striking similarities between cancer stem cells and disseminated tumor cells [55]. The DTCs treatment protection could also come from their stem-like phenotype. 

## 5. Circulating Tumor Stem Cells (CTSCs)

Not every cell in a tumor can form a metastasis. Cancer stem cells (CSCs) would seem to be the only cells able to play this role [56]. CSCs from different tumors can be identified by specific cell markers and their physiological characteristics [57]. However, most recent studies have demonstrated that some CTCs could also reseed or metastasize to distant organs [3,36,49,57].

What is the contact point between these two different cell subpopulations? Are CTCs and CSCs two different cell subpopulations, or do they join the same features, and we call them differently in different contexts?

Around 1% of the CTCs seem to have characteristics similar to stem-like cells [57,58]. In vivo experiments using circulating tumor cells isolated from patients’ blood and injected in immunodeficient animals showed that only very few cells could generate a tumor [59]. The latest results would prove that CTCs are nothing but CSCs able to withstand the bloodstream. There are two different central hypotheses about how the CTCs and the CSCs are generated in the tumor [41]. In the first hypothesis, CTCs arise as cancer stem cells, see Figure 2A.

These stem cells acquire, as a result of tumor environmental stress, peculiar characteristics that make them able to resist blood and lymphatic flow. At this stage, these cells are called circulating cancer stem cells (CTSCs).

A second hypothesis asserts that cancer stem-like cells arise from the disseminated tumor cells (DTCs), after a dormancy or hibernation period, perhaps due to some features of the dormancy itself [41], see Figure 2B.

The EMT is responsible for the generation of CTCs and it is also a critical regulator of the CSC phenotype [60]. A cell, to go through the EMT, needs to activate the EMT-inducing transcription factors (EMT-TFs). Those are classified in different protein families such as Snail, Zeb, and the basic helix-loop-helix (twisted molecules). These molecules modify the chromatin structure and are responsible for the overexpression of different molecules, such as N-cadherin, while downregulating E-cadherin. The activation of EMT, via the expression of the EMT-TFs, confers to the cells most of the CSCs proprieties [61].

The CSCs formation by EMT is a dynamic process, and it is triggered by multiple cellular signaling pathways, such as TGFβ, Wnt/β-catenin, Hedgehog, and Notch. Human CTCs isolated from murine blood become more aggressive after exposure to the hypoxic environment in vitro or in vivo [59]. Hypoxia also induces the stem markers’ overexpression, CD44, ALDH, and in breast tumors, the HER signaling is activated. The CD44 upregulation is also coordinated with the downregulation of CD24, a fundamental adhesion molecule [59]. These stem-like CTCs are very resistant to therapy. Just like the CSCs, circulating tumor stem-like cells also have different phenotypes depending on the primary tumor. A positive correlation among CD133+, level of Annexin A3 (ANXA3), and CTCs has been found for the Hepatocarcinoma CTCs [62].

The Annexin A3 is essential for the stimulation and maintenance of the stem-like cells feature. CTSCs with EPCAM- HER2+, EGFR+, NOTCH1+, and HPSE+ can generate brain metastasis in breast cancer patients [63]. The loss of urokinase plasminogen activator receptor (uPAR) and integrin β1 (intβ1) increase the dormant tumor state reducing invasiveness and metastasis potential and expanding the phenotype of the dormant disseminated tumor cells (DTCs) and CSCs [53].

Fucosylation is a reaction in which fucose is added as sugar units to a molecule. The increase of fucose sugar indicates an increase in cancer cell invasiveness, migration, and metastases formation. The CTSCs are highly fucosylated. The inhibition of the fucosylation prevents sphere formation and invasiveness of the CTSCs [53].

## 6. Circulating Tumor Cells and Radiation

Radiotherapy sometimes has the unfortunate side effect of promoting the spread of cancer cells to other organs [64,65]. The conventional photon radiotherapy is not very efficient with high-grade hypoxic tumors. A subpopulation of cells with potent tumorigenic potential, critical for regrowth and metastasis, can survive [66].

In recent years, the comprehensive study of circulating tumor cells has given rise to new studies that have cast a light on metastases formation mechanisms. Radiation can increase the incidence of metastases in several ways. Radiation could disrupt the primary tumor vasculature, which would lead to immediate shedding of CTCs. Radiation has augmented the release of viable CTCs into circulation in head and neck cancer patients with non-small cell lung cancer and squamous cell carcinoma [67]. Furthermore, radiation could induce biomolecular changes in tumor cells, favoring, for example, the notorious epithelial to mesenchymal transition, also in this case leading to an increased CTC shedding.

Additionally, radiation could produce systemic effects, such as eliminating suppressive signaling molecules by the primary tumor resulting in the proliferation of existent but previously dormant micro-metastases (DTCs) [68]. In fact, neoplastic cells might leave the primary tumor that is at an early stage of cancer development [52]. In this scenario, known as the parallel progression model, the genetic evolution that prepared the cells to become disseminated tumor cells could occur at the metastasis site, distant from the primary tumor [52]. Then, radiation could still play a key role in awakening disseminated tumor cells and forming the metastasis.

The fusion of endosomal multivesicular bodies (MVB) with the plasma membrane creates nanometric vesicles released into intercellular space, exosomes. Exosomes play a crucial role in metastasis formation and development after radiation [65]. Radiation increases the number of molecules charged into the vesicles responsible for activating the AKT pathway, which is accountable for the motility and invasiveness through the regulation of the matrix metalloproteinases (MMP). The proteinases are zinc-dependent endopeptidases and can degrade the extracellular matrix (ECM) proteins, such as collagen, laminin, and fibronectin, during cancer invasion and metastasis. After radiation, a higher amount of phosphoinositide 3K (PI3K), an enzyme involved in cellular motility and protein trafficking, has been found in the vesicles released by glioblastoma. Glioblastoma cells exposed to a medium of 4 Gy irradiated cells increase their migration and invasiveness sharply [65]. Exosomes after radiation also showed large amounts of MMP2; the fibroblast growth factor receptor 1 (FGFR1); the heat shock proteins HSP90AA1, HSP90AB1, HSP90B1; and vitronectin (VTN).

Angiostatin is implicated in the enhancement of distal metastasis, keeping the DTCs in a dormant state. After irradiation of the primary tumor, the amount of angiostatin sharply decreases. The substantial collapse of angiostatin after tumor eradication could give the DTCs the signal to start to proliferate. Furthermore, radiation is also responsible for indirectly increasing the number of CTCs, destroying the tumor vasculature, increasing hypoxia and starvation.

Photon radiation plays a role in increasing the expression of promigratory factors, such as integrins, MMP-2, MMP-9, and is responsible for activating the EMT and increasing cell motility and invasiveness. Furthermore, it enhances the Src-dependent epidermal growth factor receptor (EGFR), promoting PI3K/AKT activation and MMP-2 expression [69].

In cervical cancer cells, photon radiation enhanced cell migration via KRAS/cRAF/p38 pathway activation [70].

Photon radiation reduces the cell-cell adhesion. It induces the E-cadherin downregulation and N-cadherin upregulation, a reliable marker for the cell mesenchymal phenotype. It increases endothelial permeability and transendothelial migration. In HUVEC cells, 2 or 4 Gy of photons increased the expression and the activity of ADAM10 that, in turn, degraded VE-cadherin [71]. Breast cancer cell lines MCF7 and SKBR3 exposed to fractionated photon radiation (2 Gy x three days) increase their invasion and migration ability [66].

Furthermore, for MCF7, a substantial E-cadherin decrease and Vimentin-N-cadherin increase, after radiation, have been found. Radiation increases SLUG strongly. SLUG plays a role by sharply downregulating E-cadherin, which supports the mesenchymal phenotype by shifting the cadherin expression profile [66]. Moreover, radiation plays a role in reorganizing the actin into the cytoskeleton [72]. Migration is due to an assembly/disassembly equilibrium of the cytoplasmic microtubule and actin complex. Between 0.5 and 20 Gy, a rapid increase in actin reorganization into stress fibers have been found [72]. This is also involved in changes of the adherens junction protein, such as the VE-cadherin. These cytoskeletal changes can lead to changed cell permeability through new intercellular gap formations. A dose, ranging from 2 to 8 Gy, strongly increases cell migration. Inhibition of PI3K and mTOR is enough to block the migration completely [72]. The inhibition of AKT, which plays a crucial role in the PI3K\Akt pathway and mTOR, protein kinases, could be a promising strategy, increasing the efficacy of radiotherapy, and therefore the patient’s survival.

Cell migration and invasion seem to increase with the dose, with a maximum of around 4–5 Gy. The maximum pro-invasive effect of lung cells irradiated with 10 Gy of X-ray is 16 and 24 h after radiation [70]. After 24 h, the invasive phenotype is not evident anymore [70].

Circulating tumor cells are usually found in frequencies in the order of 1–10 CTC per mL of whole blood in metastatic disease patients. For comparison, an ml of blood contains a few million white blood cells. An acute increase of circulating cells after radiation, in a window of time consistent with changes in tumor vasculature permeability, has been found [67]. However, although these cells have a mesenchymal phenotype, they have not yet acquired the typical phenotype of CTCs, and they are not ready for the bloodstream. Most of them are destined to die in the next few minutes [67].

In any case, in the marasmus created by radiation, CTCs also manage to make room and extravasate into the blood vessels. Radiation would seem to create a fertile niche for CTCs and metastases. In BALB/c mice, the in-breast injection of a mammary carcinoma tumor cell line (4T1) after 9 Gy of half chest photon irradiation led to a higher incidence of lung metastasis and shorter survival [73,74]. Still, 2 Gy of abdomen irradiation, before colon cancer cells intravenous injection, increased the mice mortality due to higher lung metastasis. The irradiation of healthy liver tissue resulted in an increased number of in-liver metastasis after tumor cell injection [75].

In general, radiation seems to be responsible for increasing cell migration, increasing the CTCs subpopulation, and increasing the tumor’s metastatic potential. In this complex scenario, particle radiation may be more efficient for reducing CTCs, damaging and killing the hypoxic niches, and reducing the population of CSCs. Most of the damage that occurs after X-ray irradiation is through the Reactive Oxygen Species (ROS) interaction. The CSCs usually have a low level of ROS and enhanced protection from oxidative damage. The mechanism that protects CTCs is not clear, but it could be correlated with those of the CSCs. Similar to carbon ion radiation, high LET radiation could be the key to treat CSCs and cancer cells with a CTSCs phenotype before intravasation. Ion irradiation has a higher relative biological effectiveness (RBE), and therefore is effective even against the hypoxic tumor regions, resistant to conventional radiation. In addition, photon and carbon ion irradiation activate different stress gene pathways. In general, carbon ion and high LET irradiation reduce the EMT transition as compared with photon irradiation [76].

Carbon ion irradiation could also increase migration through the activation of the PI3K pathway in some cell lines, but most of the time inhibits migration and invasion.

The team of Fujita found out that a substantial reduction of the MMP-2, αvβ3, and αvβ5 integrin expressions limit the invasive potential of cells following C-ion irradiation [70].

Furthermore, carbon ion radiation reduces cell migration and invasion by reducing the guanosine triphosphatase (GTP)-bound Rac1 and GTP-bound RhoA ubiquitin-proteasome [70].

Further studies are needed to better understand the molecular mechanisms involved in generating CTCs or the molecules that give the growth or quiescence signal in DTCs. These subpopulations of cells are responsible for the formation of metastases, the primary cause of cancer death.

Ionizing radiation is also responsible for certain aspects of the generation of these cell populations. Radiation is not always able to efficiently eliminate these radio-assistant cells. These cells also have a remarkable ability to withstand any treatment, from chemotherapy to radiation. At first, we were considering only the CSCs, cancer stem cells. It was thought that this was the problem that needed to be solved to combat cancer effectively. Today it is known that CSCs cannot be considered to be the only enemy to be defeated. Rather, recent studies seem to show that, by now, we need to talk about circulating cancer stem cells.

The circulating tumor cells are difficult to find and analyze. Nevertheless, at the same time, they can give us essential information about the type of tumor, the staging, and meaningful information about the efficiency of a specific kind of treatment. Finding the right dose and type of radiation and eliminating these cell populations is fundamental, as well as understanding the molecular mechanisms that could give us, one day, an efficient weapon to use against the tumor itself.

## 7. Conclusions

CTCs are a subpopulation of tumor cells that, from the primary tumor, can intravasate into the blood and the lymphatic vessels and migrate in distant organs to form metastases [13]. Metastases are the leading cause of cancer death [1]. Radiotherapy of a multi-metastatic patient is not possible, due to the residual dose that the surrounding healthy tissues would receive in such a widespread treatment.

The road to understanding the molecular pathway involved in the formation of CTCs and their dissemination is still long. CTCs are challenging to isolate, and therefore to characterize [77]. However, the information that could come from the investigation of these cells is enormous. Therefore, it is worth the effort to find new methods to isolate and study them [78]. Characterizing these cells could give information about the stage of a tumor.

Furthermore, it could help predict a patient’s prognosis and could be used to verify a specific treatment’s efficacy. CTCs are the ideal candidates for liquid-not-invasive biopsies [74]. CTCs can migrate away from a primary tumor, but they can also resist blood flow, extravasate to specific colonization sites, and then regenerate a metastasis [13]. Once they reach the site of colonization, these cells, at this point known as disseminated tumor cells (DTCs), can remain dormant for years, even more than five years, before awakening, due to some unknown signal, and forming metastases [79].

The phenotype of CTCs is not well defined. They are single cells or clusters of hybrid cells, formed by tumor cells with stem characteristics, cells with an epithelial phenotype, cells with mesenchymal phenotype, and cells with a hybrid phenotype [80]. This last hybrid phenotype seems to be the most aggressive [80]. Is it possible to identify a specific metabolic pathway, a particular molecule, that if silenced would allow the tumor to lose its aggressiveness and metastatic potential? 

Several studies have shown that radiation, in some cases, could regress the primary tumor and the distal metastases. Today, this phenomenon is known as the abscopal effect [81]. In this case, radiation created in situ vaccine, modifying the irradiated cancer cells’ phenotype and generating a strong and not localized immune response. In this way, the immune cells could eliminate the primary tumor and the scattered distal cells. Nevertheless, radiation is a two-faced coin. Several authors have shown that it has sometimes been responsible for an increase in the number of metastases [82]. Different types of radiation may generate different responses to the formation of metastases. Some authors have shown, for example, that photon radiation, as compared with particle radiation, could increase the number of metastases [70]. However, studies about the interaction of CTCs, DTCs, and radiation are still scarce. More studies to investigate the effects of different type of radiations, dose, energy, dose/rate, on the CTCs’ formation and spread should be done.

## Figures and Tables

**Figure 1 ijms-21-09592-f001:**
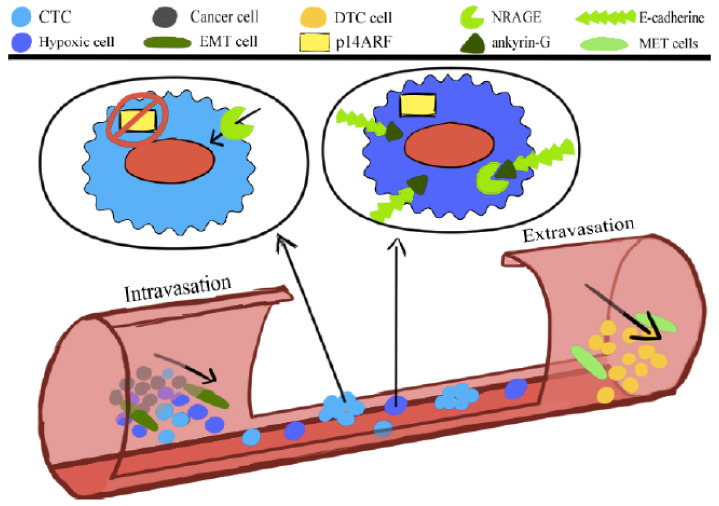
The anoikis or programmed cell death comprises the oncogenic epithelial–mesenchymal transition (EMT). The neurotrophin receptor-interacting melanoma antigen (NRAGE) molecule interacts with a component of the E-cadherin complex, ankyrin-G, and in doing so, it is blocked in the cytoplasm. Oncogenic EMT downregulates ankyrin-G, enhancing the nuclear localization of NRAGE. The oncogenic transcriptional repressor protein TBX2 interacting with NRAGE represses the tumor suppressor gene p14ARF. The product of this gene is a vital molecule to sensitize the cells to the anoikis.

**Figure 2 ijms-21-09592-f002:**
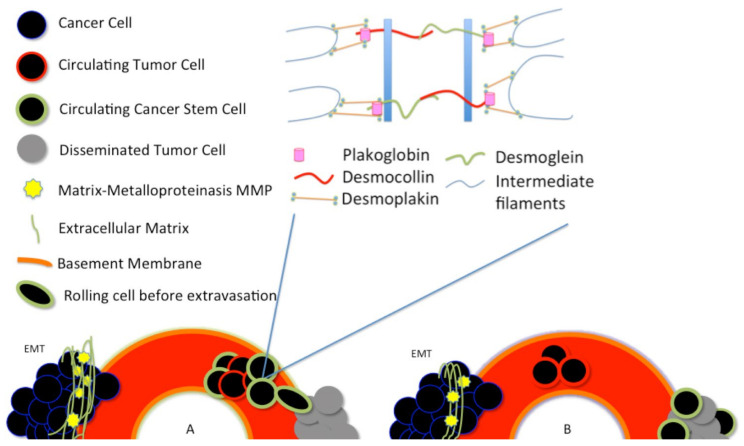
Plakoglobin is essential for the formation of circulating tumor cell clusters. Plakoglobin is the cytoplasmic component of desmosomes. There are two different central hypotheses about how the circulating tumor cells (CTCs) and the cancer stem cells (CSCs) are generated in the tumor. (**A**) In the first hypothesis, CTCs arise as cancer stem cells; (**B**) A second hypothesis asserts that cancer stem-like cells arise from the disseminated tumor cells (DTCs), after a dormancy or hibernation period, perhaps due to some features of the dormancy itself.

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
