# Peer review of "Tumor Hypoxia and Circulating Tumor Cells"

_ijms, 2020, doi:10.3390/ijms21249592_

Round 1

Reviewer 1 Report

List of suggested corrections:

  1. Please edit the title so that it will reflect the content of the manuscript more adequately.
  2. “CTCs have become a non-invasive method for studying tumor stage and tumor cell characteristics and markers [14].” – CTCs is not a name of a method, their isolation is.
  3. Where possible please cite original papers not only reviews e.g. “Similar results have also been found in pancreatic cancer [31].”
  4. “The isolation of 178 CTCs from blood vessels is a very complex operation.” – they’re not isolated from the vessel but peripheral blood.
  5. “However, most recent studies demonstrate that 308 some CTCs can also reseed or metastasize to distant organs [55].” – you are referring to multiple studies, please cite all of them (more than one, at least).
  6. Figure 2 – it’s difficult to read, requires major editing.
  7. Figure 3 – too small font size impedes reading. I would suggest using different, more contracting colours since the figure in present shape is very difficult to read.
  8. In my opinion a figure that organizes all molecular pathways concerning CTCs/DTCs/CTSCs would have been of more importance than all three attached.

Author Response

Many thanks to the reviewers for their time in proofreading our article.

We appreciate the comments and edits they asked us to make.

Below are our point-by-point responses.

Reviewer 1:

  1. Please edit the title so that it will reflect the content of the manuscript more adequately.

Authors: We have changed the title as the reviewer requested.

  1. CTCs have become a non-invasive method for studying tumor stage and tumor cell characteristics and markers [14].” – CTCs is not a name of a method, their isolation is.

Authors: Thank you. You are absolutely right. We changed the error.

  1. Where possible please cite original papers not only reviews e.g. “Similar results have also been found in pancreatic cancer [31].”

Authors: Done, thank you for your comment

  1. The isolation of 178 CTCs from blood vessels is a very complex operation.” – they’re not isolated from the vessel but peripheral blood.

Authors: Also in this case, thank you for your suggestion. Change done.

  1. However, most recent studies demonstrate that 308 some CTCs can also reseed or metastasize to distant organs [55].” – you are referring to multiple studies, please cite all of them (more than one, at least).

Authors: Done, Thank you.

  1. Figure 2 – it’s difficult to read, requires major editing.

Authors: Thank you for your comment. We did some change to the figure, now it should be more clear.

  1. Figure 3 – too small font size impedes reading. I would suggest using different, more contracting colours since the figure in present shape is very difficult to read.

Authors: Thank you very much. Also in this case we did some changes to make the figure more clear.

  1. In my opinion a figure that organizes all molecular pathways concerning CTCs/DTCs/CTSCs would have been of more importance than all three attached.

Authors: I follow your suggestion, and I merged figure 2 with figure 3. I also improved the quality of the picture. Thank you for your comment.

Reviewer 2 Report

This is an interesting review. The paper is generally well written and structured. However, in my opinion, the manuscript has some shortcomings in regards to figures that are too simple and low resolution.

Author Response

Reviewer 2:

This is an interesting review. The paper is generally well written and structured. However, in my opinion, the manuscript has some shortcomings in regards to figures that are too simple and low resolution.

Authors: Thank you for your comment. I change the figures as requested. I merged figure 2 with figure 3. Now the figure should be more complex and of better quality.